# Laparoendoscopic Rendezvous: An Effective and Safe Approach in the Management of Cholecysto-Choledocholithiasis in Selected Patients

**DOI:** 10.3390/jcm14041310

**Published:** 2025-02-16

**Authors:** Rossana Percario, Paolo Panaccio, Maria Pia Caldarella, Marco Trappoliere, Maria Marino, Maira Farrukh, Carla Di Giacomo, Giuseppe Di Martino, Giovanni De Nobili, Raffaella Marina di Renzo, Tommaso Grottola, Pierluigi Di Sebastiano, Fabio Francesco di Mola

**Affiliations:** 1General Surgery Unit, “F. Renzetti” Hospital, 66043 Lanciano, Italy; rox.percario@live.it (R.P.); paolo.panaccio@gmail.com (P.P.); g.denobili@yahoo.it (G.D.N.); 2Unit of General and Surgical Oncology, University “G. D’Annunzio” Pierangeli Clinic, Piazza Luigi Pierangeli n1, 65124 Pescara, Italy; raffaelladirenzo79@gmail.com (R.M.d.R.); tommaso.grottola@grupposynergo.com (T.G.); pierluigi.disebastiano@unich.it (P.D.S.); 3Department of Innovative Technologies in Clinical Medicine & Dentistry, University “G. D’Annunzio”, 66100 Chieti-Pescara, Italy; maira.farrukh@yahoo.it; 4Unit of Diagnostic Endoscopy and Digestive Surgery, “G. Bernabeo” Hospital, 66026 Ortona, Italy; mpiacaldarella@tiscali.it (M.P.C.); m.trappoliere@gmail.com (M.T.); maria.marino12@ymail.com (M.M.); 5Unit of Gastroenterology and Endoscopic Surgery, “F. Renzetti” Hospital, 66034 Lanciano, Italy; carla.digiacomo@als2abruzzo.it; 6Department of Medicine and Ageing Sciences, “G. D’Annunzio” University of Chieti-Pescara, 66100 Chieti, Italy; giuseppe.dimartino@unich.it; 7Department of Medical, Oral & Biotechnological Sciences, University “G. D’Annunzio”, 66100 Chieti-Pescara, Italy

**Keywords:** cholecysto-choledocholithiasis, endoscopic retrograde cholangio-pancreatography, laparoscopic–endoscopic rendezvous, common bile duct exploration, pancreatitis

## Abstract

**Background:** Different techniques have been proposed to manage Cholecysto-choledocholithiasis (CCL) advantageously in one stage. Among these, Laparoendoscopic Rendezvous (LERV) addresses the CCL issue with a laparoscopic cholecystectomy, with insertion of a guide wire into the common bile duct through an incision of the cystic duct, followed by the clearance of the bile duct carried out by the endoscopists. The aim of this study was to evaluate the safety and the efficacy of the one-stage vs. a two-stage approach (pre-operative ERCP followed by cholecystectomy), and to compare our results with data from the current literature. **Methods:** All patients that underwent LERV in our facilities between January 2018 and December 2023 were evaluated. As a control group, we included patients that underwent a two-stage technique called the “sequential approach”. The primary outcome was to evaluate the efficacy in obtaining complete clearance of the common bile duct (CBD). The secondary outcomes included morbidity, mortality, operative time, conversion rate, hospital stay and CDB stone recurrence. **Results:** 120 patients in the LERV group were included; meanwhile, 70 patients underwent pre-operative ERCP plus cholecystectomy. A 97% bile duct clearance success rate in the LERV group and 93% in the ERCP group was observed, respectively. The median intraoperative time for the one-stage technique was 122 min (*p* < 0.001) and the median hospital stay was 4 days (*p* < 0.001). In the LERV group, an overall morbidity of 15% was reported (18/120): 15 Clavien–Dindo type 1, one type 3a and two type 3b (*p* < 0.001). At a median follow-up of 14 months, five patients experienced stone recurrence. In the ERCP group, we had a 93% success rate; meanwhile, we had longer hospitalization (*p* < 0.001), 27% post-ERCP pancreatitis (*p* < 0.001) and a cumulative morbidity of 30%. **Conclusions:** LERV offers the advantages of a being single-stage procedure and shorter hospitalization, with a lower risk of clinically relevant post-ERCP pancreatitis and failed CBD cannulation.

## 1. Introduction

Stones are the most common gallbladder disease, afflicting a median of 6–22% of the population in European Countries, with higher prevalence in females (14.6%) than males (6.7%) [1]. In 80% of cases, patients will remain asymptomatic throughout their life, whereas 20% of patients with cholelithiasis are more likely to develop complications such as acute cholecystitis, acute pancreatitis, and in 5–20% of cases, choledocholithiasis [2,3]. Until about 20 years ago, the classic approach to CCL consisted of open cholecystectomy with intraoperative cholangiography, followed by open common bile duct clearance. Developments in advanced laparoscopy and operative pancreaticobiliary endoscopy changed the management of CCL [4,5,6,7], providing more options to address the issue.

Nowadays, a two-stage (the so-called “sequential”) technique combining ERCP and CBD clearance followed by delayed laparoscopic cholecystectomy represents the most common approach for CBD stones [4,8]. Also, although ERCP still remains the gold standard procedure for choledocholithiasis, it may harbour pitfalls such as a challenging cannulation or post-operative pancreatitis, stenosis of the ampulla and recurrent choledocholithiasis [9,10].

Currently, post-ERCP pancreatitis (PEP) stands out as one of the most frequent complications associated with endoscopic choledochal stone extraction, and it can lead to major systemic issues. Risk factors for PEP are well known in the literature, although additional aspects should be taken into consideration both pre and post-operatively to assess or perhaps mitigate the occurrence of PEP. An increased risk has been described in women, whereas a lower rate was observed in those who underwent CBD stone clearance by Dormia probe and EPBD (Endoscopic papillary balloon dilation) with no biliary stent. Moreover, post-ERCP, higher values of Acute phase reactants (CRP) and the Neutrophil–Lymphocyte ratio (NLR) could possibly be predictive for PEP [11]. To address CCL in one single operative session, different techniques have been proposed: laparoscopic cholecystectomy (LC) along with laparoscopic common bile duct exploration (LCBDE), LC plus trans-cystic bile duct exploration, LC plus intraoperative ERCP and LERV. In some institutions, LCBDE has become a standard as a one-stage treatment, especially when dealing with paediatric patients, allowing a potential reduction in the occurrence of post-ERCP pancreatitis, incomplete CBD stone clearance, cholangitis and gastrointestinal bleeding [12].

Laparoscopic CBD exploration (either with a trans-cystic or choledochal approach) is not currently extensively used, being a demanding technique that requires advanced laparoscopic skills and a steep learning curve [13].

In 1993, Deslandres proposed a technique that combined LC with intraoperative cholangiography (IOC) and subsequent endoscopic stone removal with ERCP in a single-stage approach, and the procedure was termed LERV [14]. Thereafter, many authors explored this method, highlighting significant improvements in the CBD clearance rate, post-operative pancreatitis occurrence and the length of stay when adopting LERV compared to the conventional two-stage approach [15]. The aim of the present work is to evaluate the role of a single-stage LERV for the treatment of CCL in a retrospective multicentric study compared to the conventional two stage approach (ERCP followed by LC), with special attention to post-operative complications and hospital stays.

## 2. Methods

We conducted a retrospective observational review of clinical cases of CCL observed in General Surgery Units of the “F. Renzetti” Hospital in Lanciano (Chieti, Italy) and the Unit of General and Surgical Oncology of the Pierangeli Clinic (Pescara, Italy), both affiliated with the University G. D’Annunzio, Chieti-Pescara (Italy).

The members of the surgical and endoscopic teams involved have performed at least 30 LERVs and 50 ERCP procedures, respectively, having almost 10 years of experience in complex hepato-pancreaticobiliary procedures.

From January 2018 to December 2023, 1554 consecutive LCs for gallstones were performed in the two above-mentioned facilities. A total of 190 patients with cholecysto-choledocholithiasis were identified and subdivided as follows: 120 patients were treated with LERV (Group I) and 70 patients were treated with ERCP (Group II). When the preoperative work-up was indicative of concomitant cholecysto-choledocholithiasis, a single-stage approach was routinely planned, whether in an elective or urgent setting, following a multidisciplinary meeting with dedicated endoscopists, surgeons and radiologists. The diagnosis was based on clinical and radiological features. Inclusion criteria for proposing a LERV approach were: altered liver function tests and typical findings of CCL at ultrasound (CBD diameter greater than 10 mm or direct visualization of stones with a diameter of between 2 and 15 mm), CT scan and/or magnetic resonance cholangiography (cMRI) or failure of a previous ERCP. Exclusion criteria for LERV were a clinical diagnosis of acute cholangitis with septic shock, impacted choledochal stones (diameter of stones > 1.5 cm), jaundice with total bilirubin levels greater than 4 mg/dL or suspected malignant stenosis. The latter group of patients was scheduled for pre-operative ERCP and included in our control group. The cut-off value of bilirubin was borrowed from a previous study where we reported that patients with values above 4 gr/dL have an increased risk of postoperative complications [16].

As a summary, a diagnostic and therapeutic flowchart is reported in Figure 1.

The collection of demographic data included age, gender, BMI and ASA (American Society of Anaesthesiologists) score. The Clavien–Dindo score system of complications after surgery was applied, inclusive of post-procedural pancreatitis, perforation, bleeding, intra-abdominal or port-site infections, bile leak, cholangitis, residual choledocholithiasis and thromboembolism [17].

All procedures were performed in accordance with good clinical practice and in compliance with the guidelines of the Helsinki Declaration for studies using human subjects. Institutional Review Board (IRB) approval was obtained, although not necessary for the study’s purposes (registered on 8 January 2025, no. 01/2025). No experimental processes requiring further approvals were applied, this study being structured as a retrospective evaluation, and the choices of treatment were accordant with current protocols. Permission to use clinical data for research and publication purposes was obtained at the time of consent for the procedure. Septic patients with cholangitis or cholecystitis were treated according to the Tokyo Guidelines [18]. Post-ERCP pancreatitis (PEP) was defined in accordance with the 2020 ESGE guidelines on ERCP-related adverse events, associated with new or worsened abdominal pain combined with elevated pancreatic enzymes (amylase or lipase ≥3 times upper limit of the normal) [19].

### 2.1. Outcomes of the Study

The primary outcome reported is the efficiency of CBD complete clearance. The secondary outcomes assessed morbidity, mortality, operative time, conversion rate, in-hospital stay and late recurrence of CBD stones.

### 2.2. Technical Details: Single Stage Approach—Laparoendoscopic Rendezvous

LC was performed with the standard 4-port technique; the first surgeon operating between the patient’s legs and the assistant on the left side (French position). After a careful identification of Calot’s triangle, dissection and division of the cystic artery was carried out. This step was then followed by incision of the cystic duct and insertion of a 5 Fr urological catheter (Bracci-catheter) through the cut for intra-operative cholangiography using an Olsen clamp, as reported in Figure 2A. After completing the cholangiography, a guidewire with a hydrophilic tip (Jagwire™, Boston Scientific©, Marlborough, MA, USA) was advanced through the catheter, sticking out through the sphincter of Oddi and papilla in the duodenum, as reported in Figure 2B. Subsequently, the endoscopist identified the tip of the wire through the scope, grabbed it with a snare and carefully it pulled out of patient’s oral cavity. A sphincterotome was introduced along the guide, allowing the selective cannulation of the papilla. Choledochal cannulation was followed by a sphincterotomy or EPBD (Endoscopic papillary balloon dilation) in more demanding cases. After CBD exploration, the stones were removed using a Fogarty angioplasty catheter and/or Dormia basket in cases of larger calculi, as reported in Figure 2C,D. Complete removal of the obstruction was finally confirmed by endoscopic cholangiography. If incomplete clearance or intraprocedural bleeding was suspected or encountered, a biliary plastic or uncovered metallic stent was introduced into the CBD. After the endoscopy, the laparoscopic procedure was completed by clipping the cystic duct, dissecting the gallbladder from the liver bed and placing it into a retrieval bag for extraction. Before wound closure, intra-abdominal drainage was placed in all cases.

### 2.3. Two-Stage Approach—Pre-Operative ERCP Followed by LC: Technical Details

Under general anaesthesia, the papilla was reached with an endoscope. After cannulation and CBD exploration, a Fogarty angioplasty catheter (Edwards Life Sciences Irvine, Santa Ana, CA 92711-1150, USA) and/or Dormia basket were used to perform the extraction—as shown in Figure 2C,D—proceeding with sphincterotomy or EPBD balloon dilation when required. Lastly, endoscopic cholangiography was conducted to assess the removal of the obstruction and followed by the placement of a biliary plastic or uncovered metallic in CBD in cases of bleeding or incomplete clearing. A conventional LC followed the pre-operative ERCP after a median of 4 days (2–21), as described in detail below.

### 2.4. Pre- and Post-Operative Management

No differences were reported in perioperative management between the two groups.

No mechanical bowel preparation was required and single-shot antibiotic therapy with cephazolin 2 gr i.v. was administered to all patients. Postoperative care was alike in both groups. All patients received thrombotic prophylaxis with low molecular-weight heparin 12 h before surgery and then once daily until discharge, according to the patient’s comorbidities. Proton pump inhibitors, analgesics and rectal indomethacin were routinely administered. Nasogastric tube was removed on awakening; the urinary catheter was placed after induction of general anaesthesia in patients with a known history of prostatic hypertrophy or with cardiovascular pathologies requiring monitoring. Intra-abdominal drainage was routinely placed, and removed on the first post-operative day if no complications or bile leakage occurred. Criteria for discharge included the absence of symptoms, flatus passage, acceptable feeding and good pain control. After discharge, patients were followed up in our outpatient clinic ten days after surgery for control and suture removal. A telephone follow-up (TFU) after hospital discharge was conducted for all patients. A median follow-up of 14 months (range 3–59) is reported.

### 2.5. Statistical Analysis

Continuous variables were expressed as the median (ranges) due to a non-normal distribution. Categorical variables are expressed as frequency and percentage. Qualitative variables were compared with Pearson’s Chi-Square test or Fisher’s Exact test as appropriate. Continuous variables were compared with Mann–Whitney U tests. Continuous variables were tested for normal distribution with the Shapiro–Wilk test. All tests were considered significant for a *p*-value less than 0.05. Statistical analysis was performed with Stata Software v18 (Stata Corp., Station, TX, USA).

## 3. Results

### 3.1. One-Stage Approach: LERV Group

A total of 120 patients presenting with CCL were treated with LERV between January 2018 and December 2023. Emergency cases were medically treated and scheduled for an elective procedure during the same hospitalisation, according to the Tokyo guidelines [18]. Preoperative assessment of patients with confirmed or suspected CCL was carried out according to European Society of Gastroenterology (ESGE) [5,6,7,8,9,10,11,12,13,14,15,16,17,18,19] indications. Previous abdominal surgery was not an exclusion criterion for a laparoscopic approach. Eight patients sustained a previous Billroth-2 partial gastrectomy, and in two of these cases, a previous failed ERCP was reported due to difficult CBD cannulation. Demographical data and post-operative characteristics are summarised in Table 1.

The median operative time was 122 (range 95–220) min. The median hospital stay was 4 days (range 2–38). No post-operative mortality was observed.

The overall morbidity was 15% (18/120 pts). In detail, as reported in Table 2, 12 mild post-operative pancreatitis and one surgical site infection occurred (Clavien–Dindo I). In one case (Clavien–Dindo IIIa), a patient who experienced mild pancreatitis required prolonged antibiotics treatment due to concomitant cholangitis. In all cases, the post-operative pancreatitis was considered transitory and not clinically relevant. Two severe surgical complications occurred, both requiring a laparoscopic re-operation: in one case, postoperative intraperitoneal bleeding was reported due to wide adhesiolysis; in the other case, a bile leak due to unrecognised Luschka duct led to an intra-abdominal abscess and biliary collection. Bleeding after endoscopic sphincterotomy occurred in four cases, requiring blood transfusions in three cases and endoscopic haemostasis in one case. Conversion to open surgery was necessary in two patients: one patient presented with adhesions after partial gastrectomy and the other for considerable bowel distension after ERCP that prevented a safe completion of the LC.

In all patients, CBD cannulation with a hydrophilic tip jag wire through the cystic duct was carried out; 113 patients (94%) required preventive biliary sphincterotomy (EST). In seven patients (6%), EPBD (Endoscopic papillary balloon dilation) was performed due to a giant periampullary diverticulum (four cases) or Billroth II gastrojejunostomy (three cases) that precluded a swift cannulation.

A successful duct clearance was attained in 97%, whereas four patients needed additional ERCP during the same hospitalization.

Eleven patients (9%) required a biliary plastic stent positioning for preventive purposes (intraoperative papillary bleeding or suspected impacted stones/incomplete clearance). In all cases, the successive biliary stent removal and cholangiography showed no persistent CBD stones. At a median follow-up of 14 months (range 3–59), five patients experienced stone recurrence, requiring in two cases additional ERCP. In the three cases that did not require a redo of the ERCP, the ultrasound findings for choledocholithiasis were not confirmed with magnetic resonance cholangiopancreatography.

### 3.2. Two-Stage Approach: Pre-Operative ERCP Followed by LC

Patients with jaundice and bilirubin levels greater than 4 mg/dL (nv 0.1 to 1.2 mg/dL) were scheduled for a sequential approach with ERCP followed by laparoscopic cholecystectomy. Interval times were established in accordance with bilirubin levels, the patient’s comorbidities and the post-ERCP course. A total of 70 patients were included in this group. Preoperative assessments were carried out as in the LERV group. Demographical data, patients’ characteristics and post-operative findings are summarised in Table 1.

The required median time for the endoscopic procedure was 42 (range 24–105) min. The median hospital stay was 10 days (range 5–38). No post-operative mortality was observed. The overall morbidity was 30% (21/70).

In detail, as reported in Table 2, 13 patients experienced clinically non-relevant pancreatitis. However, six cases of clinically significant post-operative pancreatitis and three cases of cholangitis (Clavien–Dindo IIIb) were reported. In one patient with severe pancreatitis, intensive care treatment and a prolonged hospital stay was needed, although no surgery was required. Eleven patients (16%) presented with bleeding from the ampulla after ERCP. In all cases, blood transfusions with additional ERCP successfully controlled the bleeding source. During ERCP, two periampullary duodenal perforations occurred and were managed by placing a metallic duodenal stent and with conservative treatment.

A successful duct clearance was accomplished in 93% of patients. In two out of five patients with residual choledocolitiasis, additional ERCP was necessary during the same hospitalization due to the unsuccessful cannulation of the CBD with a hydrophilic tip jag wire, whereas in another case, laparoscopic CBD exploration (LCBDE) was used to successfully treat the patient (Figure 3).

In all cases, laparoscopic cholecystectomy was performed during the same hospitalization. In compliance with bilirubin values and patient comorbidities, a median of 4 days’ (2–21) time after ERCP was observed.

In contrast with the LERV group, 30% of the patients required a biliary stent, while in one case, multiple stents were necessary due to manifold stones. The subsequent stent removal and cholangiography showed no persistence of CBD stones. At a median follow-up of 14 months (range 3–59), seven (10%) patients experienced stone recurrence.

## 4. Discussion

CCL implies the concomitant presence of stones in the gallbladder and the common bile duct. Most patients do not show any symptoms for a long time, and over a 15 year-long follow-up, only 25% of patients will develop overt disease.

Surgical or endoscopic CBD clearance with simultaneous (one-stage technique) or delayed LC (two-stage technique) are the strategies of choice for CCL management, but according to the current literature, there is no agreement yet to declare which the most successful technique. A recent retrospective multicentre cohort study comprising almost 600 patients demonstrated that in the multivariate analyses, intervention (one-stage), number of CBDS ≥ 2, biliary drainage, the use of a mechanical lithotripter and gallbladder stones were identified as significant factors associated with the recurrence of CBD stones [20] in the multivariate analyses.

A single procedure is definitely convenient for the patient, and it means that during one hospital stay, CCL can be thoroughly addressed. Among the one-stage procedures available, the LERV technique combines laparoscopic and endoscopic treatments of CCL [14], but in the last 20 years, the evolution of laparoscopic surgery and the expanded skills of some surgeons—especially in Eastern Asia countries—have brought up an another feasible one-stage option in the treatment of CCL: laparoscopic common bile duct exploration (LCBDE). Many studies comparing LCBDE with LERV or two-stage techniques showed that the first is associated with a shorter in-hospital stay, lower post-operative pancreatitis, higher CBD stone clearance rates and higher post-operative bile leakage rates. One limitation to the diffusion of the LCBDE is its logistic feasibility in clinical practice, as the technique requires expertise and advanced laparoscopic skills. Among European countries, some authors have reported in their metanalyses that in Germany, two-stage management was the preferred method in 99% of patients with suspected CBD stones, and there was a conversion rate of 43% in the patients when LCBDE was attempted. On the other hand, in the UK, about 61% of upper-gastrointestinal or hepatopancreaticobiliary surgeons performed LCBDE when a single-stage technique was pursued, 25% demanding postoperative ERCP and 13% performing either LCBDE or ERCP when they encountered CBD stones [20,21]. Nevertheless, the trend over the last decade has not progressed much towards LCBDE utilisation in favour of endoscopic (two-stage sequential approach) or LERV procedures, resulting in decreased familiarity with LCBDE by senior surgeons and their trainees [13,22,23,24,25]. In fact, in the present study, we reported only one case treated with LCBDE.

Compared to the two-stage approach (LC plus ERCP), LERV can be associated with higher efficacy and a better safety profile [26,27,28,29]. In one of the most significant studies, Ricci et al. are proponents of the combined LC and intraoperative ERCP approach (LERV), outlining it as the safest and most successful choice [25]. Regarding the post-ERCP pancreatitis rate, the literature reports that it ranges between 1.6 to 15.1%, meaning that the widely adopted sequential approach (pre-ERCP plus LC) is less beneficial [20,25].

However, not all authors of the contemporary literature confirm the superiority of the single-stage approach compared to the two-stage approach [6,30,31,32,33].

Our experience has shown results in line with those reported by the current literature, acknowledging that LERV is effective, allowing CBD clearance in 116 cases (97% success rate), while this result was achieved in 93% of the patients treated with the two-stage approach. In all cases, successful cannulation of CBD was accomplished, although four patients in the LERV group required additional ERCP for incomplete CBD clearance due to impacted stones. The 30% rate of stent placement during the sequential approach supports the difficulty of achieving complete clearance of the biliary tract. Furthermore, this aspect is even more interesting considering that eight patients had a surgically altered anatomy (previous Billroth-2 partial gastrectomy), and in two of these cases, a previous ERCP attempt failed due to difficult CBD cannulation. During LERV, a higher rate of selective CBD cannulation and the avoidance of a high-pressure injection of the contrast medium into the pancreatic duct are directly related to a lower rate of post-ERCP pancreatitis [33,34,35]. In the single-stage approach, preventing additional ERCP after failed biliary cannulation allows lower complication rates and patient discomfort [36,37].

Our series showed an interesting safety profile, as the overall morbidity reached 15%, while in the ERCP group, this was 30%—mainly due to post-operative pancreatitis. As reported in Table 2, only 2.5% of patients experienced severe complications requiring intervention or reoperation (Clavien–Dindo IIIa or IIIb) in the LERV group, compared to 16% (nine IIIb and two IV) of the control group. It is interesting to point out that patients who experienced complications in the LERV group had mainly biochemical post-operative pancreatitis, which did not affect the post-operative course.

Confirming the effectiveness of LERV for CBD clearance, the subgroup of patients (9%) that required a preventive positioning of biliary plastic stents during LERV ended up with stent removal and also received verification, through cholangiography, of the non-persistence of CBD stones. Moreover, at a median follow-up of 14 months (range 3–59), only five patients experienced stone recurrence, requiring in two cases supplementary ERCP. In comparison with the LERV results, in the ERCP group, 30% of the patients required a biliary plastic stent, and in three cases, even multiple stents positioned due to the presence of stacked CBD stones. A plausible explanation for this may reside in the inclusion criterion for the sequential therapy, which comprised stacked gallstone jaundice patients, therefore representing a bias in our study.

A drawback of the LERV technique includes the problem of bowel distension due to endoscopic gas insufflation during ERCP, which should be minimized or aspirated as much as possible at the conclusion of the endoscopic phase. Another possible pitfall could arise in special cases such as Mirizzi syndrome, which might require traditional catheterization combined with surgical common bile duct exploration to complete the operation. Additionally, many studies have stated that LERV is associated with a shorter hospital stay when compared with traditional two-stage treatment [25,38,39,40,41,42,43]. These data are confirmed by our experience, with a median of 4 (2–38) days’ hospital stay. Also, the two-stage approach presents the disadvantage of the waiting period required to ensure that no post-ERCP complications have occurred before proceeding to cholecystectomy, with an interval time between the two procedures that, in our study, reached a median of 4 days (range 2–21)—thus contributing to a lack of compliance and longer hospital stays for patients, with a significant increase in costs.

Despite the numerous *pros* of the LERV, the two-stage approach (pre-operative ERCP followed by LC) remains the treatment of choice in many facilities [8,20,21,25,30,31,44]. The feasibility of one-stage procedures such as LERV is not straightforward; they require good operating room organisation and the coordination of dedicated surgical and endoscopic teams—also in emergency settings.

Therefore, is not unforeseen that separately scheduling the two teams seem easier to manage. Moreover, the shortage of skilled biliary endoscopists and the difficulties or even the lack of cooperation between endoscopic and surgical teams pushes this decision towards the two-stage technique [28,45]. In our work, we endeavoured to manage and organise the operating rooms and the teams in the best way possible to achieve close collaboration and improved outcomes in the long run [46]. Besides this, it is important to underline that LERV needs only basic and cheap laparoscopic equipment in addition to the ability to perform an intraoperative cholangiography [29,47,48].

LERV also allows one to address the issue during a single hospital admission and with a single administration of general anaesthesia, with greater acceptance by patients as well as lower risks. Current evidence demonstrates that LERV has shorter hospital stays and fewer complications compared to two-stage approaches, even in emergency settings [27,28,49,50]. The limitations of the study reside in its retrospective design and the relatively limited sample size—especially for the ERCP group. Furthermore, another limitation is represented by the exclusion criterion of the presence of clear jaundice for the LERV group, restricting them to the two-stage approach group.

Although the advantages of the LERV are undeniable, we reckon that a careful selection of patients with CCL must be undertaken prior to engaging the technique, and future prospective, multicentre randomized controlled study should be conducted.

## 5. Conclusions

Choledocholithiasis can be found in almost 5–20% of patients undergoing cholecystectomy. Even to this day, the debate concerning the ideal management of CBD stones remains open. Current evidence has shown that single-stage procedures in CBD stone management are the most efficient and cost-effective. Several studies have shown that LERV enhances patients’ compliance, with high success rates over one convenient, short hospital stay and lower costs. The data we collected in our experience confirm that the LERV technique is a safe procedure in selected cases and allows for good clinical outcomes, with a lower incidence of clinically relevant post-operative ERCP pancreatitis.

## Figures and Tables

**Figure 1 jcm-14-01310-f001:**
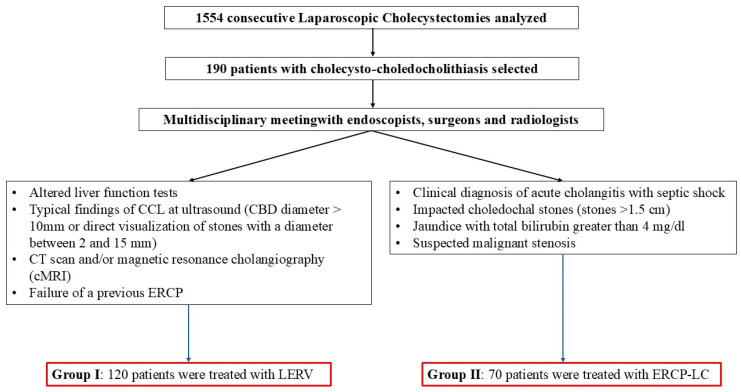
Diagnostic and therapeutic flowchart.

**Figure 2 jcm-14-01310-f002:**
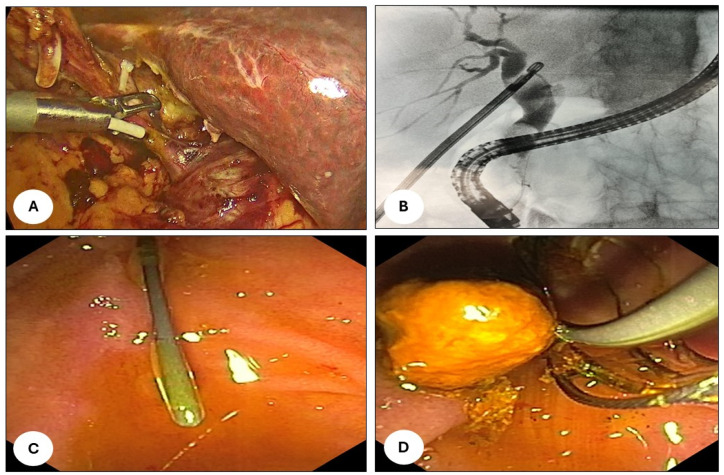
(**A**) Identification of Calot’s triangle after the division of the cystic artery. Incision of the cystic duct and insertion of a 5 Fr urological catheter (Bracci-catheter) through the cut for intra-operative cholangiography using an Olsen clamp (Karl Storz, 78532 Tuttlingen, Germany). (**B**) Cholangiography demonstrated a large choledocal stone. A guidewire with a hydrophilic tip was passed through the papilla. (**C**) The tip of the wire was identified through the scope, grabbed with a snare and carefully pulled out of the patient’s oral cavity. (**D**) After CBD cannulation and exploration, the stones were removed using a Fogarty angioplasty catheter and/or Dormia basket in cases of larger calculi.

**Figure 3 jcm-14-01310-f003:**
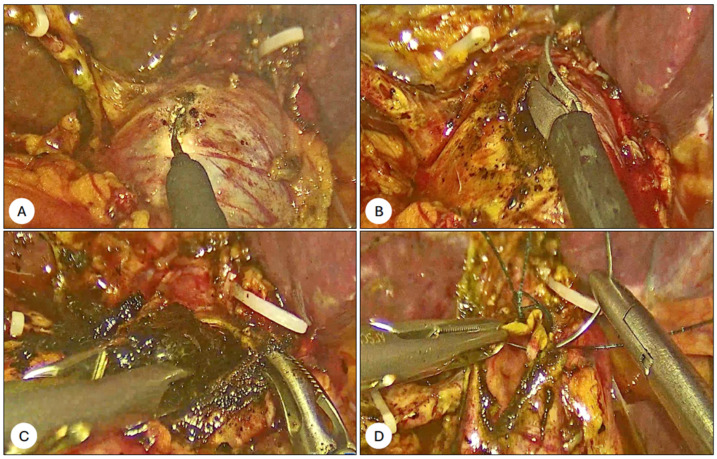
Intraoperative findings in the case of laparoscopic commom bile duct exploration (LCBDE). (**A**). Common Bile Duct (CBD) enlarged. (**B**). Opening CBD for clearance. (**C**) Clearance of CBD. (**D**) Suturing CBD with 3/0 barbed suture.

**Table 1 jcm-14-01310-t001:** Demographical and post-operative data of the LERV group and control group (sequential approach).

	Laparoendoscopic Rendezvous Group	ERCP Group	*p*-Value
Patients, *n*%	120	70	
Sex (M/F), *n*%	51/69	25/45	0.357
ASA Score, *n*%	10 ASA I (8.3%)	10 ASA I (14%)	0.001
	41 ASA II (34.2%)	11 ASA II (16%)	
	66 ASA III (55%)	39 ASA III (56%)	
	3 ASA IV (2.5%)	10 ASA IV (14%)	
BMI status (kg/cm)	26.3 (24.4–28.1)	26.8 (24.5–28.3)	0.433 °
Bile duct clearance, *n*%	116 (97%)	65 (93%)	0.117
Residual choledocholithiasis, *n*%	4 (3.3%)	5 (7%)	0.233
Redo ERCP, *n*%	4 (3.3%)	3 (4%)	0.737
Median operative time (min median IRQ)	122 (95–220)	65 (48–120)	<0.001 °
Median hospital stay (days median IRQ)	4 (2–38)	10 (5–38)	<0.001 °
Time distance to laparoscopic cholecystectomy (day median IRQ)	-	4 (2–21)	NA
Stent placement during ERCP, *n*%	4 (0.03%)	21 (30%)	<0.001 °
Types of stents used, *n*%	Plastic, n. 4 (100%)	Plastic, n. 18 (86%) Metallic, n. 3 (14%)	
Need for multiple stents, *n*%	0 (0%)	3/21 (14%)	<0.001 °
Stone recurrence at 14 (3–59) months follow-up, *n*%	5 (4%)	7 (10%)	0.135 *

* Fisher exact test. ° Mann–Whitney U test. ERCP: Endoscopic retrograde cholangiopancreatography, ASA: American Society of Anaesthesiologists. NA = not applicable.

**Table 2 jcm-14-01310-t002:** Post-operative complications of the LERV group and control group (sequential approach).

	Laparoendoscopic Rendezvous Group	ERCP Group	*p*-Value
Patients, *n*%	120	70	
Overall morbidity, *n*%	18/120 (15%)	21/70 (30%)	<0.001 *
Clavien–Dindo I, *n*%	15 (12.5%)	13 (18.5%)	
Clavien–Dindo IIIa, *n*%	1 (0.8%)	0	
Clavien–Dindo IIIb, *n*%	2 IIIb grade (1.6%)	9 IIIb (13%)	
Clavien–Dindo IV, *n*%	0 (0%)	2 (3%)	
PEP, *n*%	10 (8.3%)	19 (27%)	<0.001 *
Perforation, *n*%	0 (0%)	2 (3%)	0.134 *
Bleeding after ERCP, *n*%	4 (3.3%)	11 (16%)	0.004 *
Cholangitis, *n*%	1 (0.01%)	9 (13%)	<0.001 *

* Fisher exact test. ERCP: Endoscopic retrograde cholangiopancreatography, PEP: Post-ERCP pancreatitis.

## Data Availability

The original contributions presented in the study are included in the article.

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
