# Peer review of "Laparoendoscopic Rendezvous: An Effective and Safe Approach in the Management of Cholecysto-Choledocholithiasis in Selected Patients"

_jcm, 2025, doi:10.3390/jcm14041310_

Round 1
Reviewer 1 Report
Comments and Suggestions for Authors
The authors evaluate the efficacy and safety of the laparoendoscopic rendez-vous procedure (LERV) compared with a conventional two-stage approach (preoperative ERCP plus cholecystectomy) for cholecyto- choledocholithiasis in selected patients. In addition, they compare the results obtained with those published in the literature. They conclude that the LERV procedure offers the advantage of a single-stage procedure and a shorter hospital stay with a lower risk of clinically relevant post-ERCP pancreatitis and failed CBD cannulation.
I read this paper with great interest. Compared to a previous submission (manuscript ID: jcm-3040850), which was rejected in the same journal due to the lack of a comparison group and therefore low scientific quality, the authors have made additional efforts and included a comparison group. In its current form, the manuscript has been considerably improved and now provides valuable information on current pathology. Nevertheless, I have some comments and suggestions for improvement:
1. Abstract – Since the authors included a comparison group, they should provide p-values for each comparative outcome between groups (e.g., hospitalization, success rates, number of complications, overall morbidity rates, etc.). Please update.
2. The authors eventually analyzed 190 (120+70) patients with coledocholithiasis, but they failed to describe the groups in the methodology. The authors should clearly describe that 120 patients were treated with LERV (group I) and 70 patients were treated with ERCP (group II).
3. In addition to the previous comment, the authors should state how the patients were selected for each procedure. Who decided on the procedure and what were the criteria for LERV or ERCP treatment.
4. For the exclusion criteria, the authors state that patients with total bilirubin >4 mg/dl were excluded and selected for ERCP. This requires a detailed explanation. In my opinion, if malignancy was excluded, there is no contradiction with LERV. This could be a source of bias and therefore needs to be adequatelly adressed.
5. Although this is a retrospective study, the authors should obtain approval from their Institutional Review Board for data processing. Even if the IRB has decided that ethics committee approval is not required, the date and number of this decision should be provided.
6. The baseline characteristics (demographics) of the patients should be presented in more detail to verify that the two comparison groups were symmetrical. I would suggest reporting mean or median weight, height, BMI and comorbidities.
7. Now the authors have presented the comparison group. I would suggest adding a flow chart of the study. I suggest using a PRISMA flow chart template.
8. The authors state that all continuous variables were expressed as median (ranges) due to non-normal distribution, but they do not state which statistical test was used to test the normality of the data distribution.
9. Table 1 – Any abbreviation appearing in the table should be listed in the legend below the table (e.g. ASA, ERCP. In addition, the authors should indicate next to each variable how the variable was expressed (as n (%) or median (IQR)).
10. Several p-values in Table 1 have been labeled with '°'. Please indicate the meaning of this symbol in the table legend.
11. For better clarity, Figures 1 – 4 should be combined into one figure. (Figure 1 – A,B,C,D). The same as it was dane for Figure 5, please revise.
12. The complication in relation to Clavien – Dindo should be presented more clearly and in a more reader-friendly way. For example, stages I and II should be presented separately. I would suggest creating a separate table for complications related to Clavien – Dindo classification and for each stage, provide the number of complications in both groups along with the p-values.
13. The discussion is generally well written, but lacks a critical discussion of conflicting studies, particularly those advocating a two-stage approach. This would strengthen the contextual validity of the results.
14. The conclusions should be presented in a separate paragraph. After the discussion, the authors should add the paragraph "5. Conclusions", as is customary in the journal.
15. The quality of the English language is satisfactory, but needs some improvement in terms of spelling and grammatical errors. The manuscript should be revised by a native English speaker or a professional language editor to improve grammar and readability.
16. iThenticate report showed the amount of 30% of wording duplication in the manuscript which is relatively high for academic journal and should be reduced to <20%.
Comments on the Quality of English LanguageThe quality of the English language is satisfactory, but needs some improvement in terms of spelling and grammatical errors. The manuscript should be revised by a native English speaker or a professional language editor to improve grammar and readability.
Author Response
- Abstract – Since the authors included a comparison group, they should provide p-values for each comparative outcome between groups (e.g., hospitalization, success rates, number of complications, overall morbidity rates, etc.). Please update.
Comment: Thanks for the comment. We agree with reviewer #1 and in the revised Abstract section we have added the requested information
- The authors eventually analyzed 190 (120+70) patients with coledocholithiasis, but they failed to describe the groups in the methodology. The authors should clearly describe that 120 patients were treated with LERV (group I) and 70 patients were treated with ERCP (group II).
Comment: Thanks for the comment. We agree with reviewer #1 and in the revised methods section (lines 141-144) we have specified how the patients were divided
- In addition to the previous comment, the authors should state how the patients were selected for each procedure. Who decided on the procedure and what were the criteria for LERV or ERCP treatment.
Comment: Thanks for the comment. Indeed, in the methods section from line 144 to 160 it is clearly written how the decision-making process occurred for each individual patient included. However, in the revised version we have better specified this point (see also point 7). Furthermore a PRISMA flow chart was added
- For the exclusion criteria, the authors state that patients with total bilirubin >4 mg/dl were excluded and selected for ERCP. This requires a detailed explanation. In my opinion, if malignancy was excluded, there is no contradiction with LERV. This could be a source of bias and therefore needs to be adequately addressed.
Comment: Thanks for the comment. The cut-off of bilirubin values comes from a previous study where we reported that patients with values above 4 gr/dl have an increased risk of postoperative complications. In the revised version we clarified this point and added a specific reference (16) (lines 157-159)
- Although this is a retrospective study, the authors should obtain approval from their Institutional Review Board for data processing. Even if the IRB has decided that ethics committee approval is not required, the date and number of this decision should be provided.
Comment: Thanks for the comment. In the revised version we have added the requested information
- The baseline characteristics (demographics) of the patients should be presented in more detail to verify that the two comparison groups were symmetrical. I would suggest reporting mean or median weight, height, BMI and comorbidities.
Comment: Thanks for the comment. We disagree with reviewer#1. In fact, the purpose of the study was to evaluate the efficacy of single versus consecutive approach in terms of post ERCP pancreatitis, failed CBD cannulation and hospital stay (and indirectly on management costs). These parameters are not related to patients' comorbidities but to the procedure itself. In addition, having specified inclusion and exclusion criteria it is not possible to have homogeneous data as requested by reviewer. A randomized, not retrospective study should be performed in such a case. In summary, adding such data would not help the reader but rather may confuse ideas and take the reader out of the main focus of our paper
- Now the authors have presented the comparison group. I would suggest adding a flow chart of the study. I suggest using a PRISMA flow chart template.
Comment: Thanks for the comment. In the revised version we have added a PRISMA flow chart named Table 1 as reviewer # 1 suggested.
- The authors state that all continuous variables were expressed as median (ranges) due to non-normal distribution, but they do not state which statistical test was used to test the normality of the data distribution.
Comment: Thanks for the comment. Continuous variables were tested for normal distribution with Shapiro Wilk’s test (chapter 2.5. Statistical Analysis (lines 290-291)
- Table 1 – Any abbreviation appearing in the table should be listed in the legend below the table (e.g. ASA, ERCP. In addition, the authors should indicate next to each variable how the variable was expressed (as n (%) or median (IQR)).
Comment: Thanks for the comment. In the revised version we have added such information
- Several p-values in Table 1 have been labeled with '°'. Please indicate the meaning of this symbol in the table legend
Comment: Thanks for the comment. In the revised version this point is clarified (line 311-313) in the ex-Table 1 now renamed Table 2
- For better clarity, Figures 1 – 4 should be combined into one figure. (Figure 1 – A, B, C, D). The same as it was done for Figure 5, please revise.
Comment: Thanks for the comment. In the revised version there exist now only figure 1 A, B, C, D
- The complication in relation to Clavien – Dindo should be presented more clearly and in a more reader-friendly way. For example, stages I and II should be presented separately. I would suggest creating a separate table for complications related to Clavien – Dindo classification and for each stage, provide the number of complications in both groups along with the p-values.
Comment: Thanks for the comment. We added a “Table 3” in which the complications observed in the two different procedures are described in detail. As suggested, Table 1 reports a PRISMA flowchart. Patient’s data and other post operative characteristics are reported in Table 2.
- The discussion is generally well written, but lacks critical discussion of conflicting studies, particularly those advocating a two-stage approach. This would strengthen the contextual validity of the results.
Comment: Thanks for the comment. We agree with the reviewer’s suggestion and in the revised version we add critical discussion of conflicting studies.
- The conclusions should be presented in a separate paragraph. After the discussion, the authors should add the paragraph 5. Conclusions", as is customary in the journal.
Comment: Thanks for the comment. In the revised version is done what suggested
- The quality of the English language is satisfactory, but needs some improvement in terms of spelling and grammatical errors. The manuscript should be revised by a native English speaker or a professional language editor to improve grammar and readability.
Comment: Thanks for the comment. The current manuscript version was revised by a native English speaker as suggested (Dr. M. Farrukh)
- iThenticate report showed the amount of 30% of wording duplication in the manuscript which is relatively high for academic journal and should be reduced to <20%.
Comment: Thanks for the comment. We have carried out a new check with the software Turnitin ( https://www.turnitin.com/products/ ) obtaining a result of 21% of wording duplication. We have provided to modify some sentences to obtain a result < 20%

Reviewer 2 Report
Comments and Suggestions for Authors
Dear All,
I was pleased to review the article Laparoendoscopic Rendez-Vous: An Effective and Safe 2 Approach in the Management of Cholecysto-Choledocholithiasis in Selected Patients
I read this article carefully and with great interest, the treatment of cholecysto-choledocholithiasis is still a big challenge and, sometime, is difficult to make the good decision. The risk of developing acute pancreatitis or severe angiocolitis can greatly complicate the patient's prognosis, and the equipment and the availability of specialists also playing an important role.
In my opinion the content of the manuscript and the direction are clear but I have some suggests:
- The title expresses clearly the content of the manuscript.
- The abstract is a clear summary of the aim. There are abbreviations that are not explained on first appearance - for example, line 30 - CBD or line 32 – CDB. Also, there are situations in the text - line 92 for example - when an abbreviation is explained again (in this case, LERV) or line 177, or the abbreviation is given and then explained - for example, line 77 - EPBD (Endoscopic papillary balloon dilation). In the Abstract I think that instead of Discussion (line 40), it should be Conclusions of this study.
- The introduction section should summarize the current state of the topic as well as clearly define the aim of the study. In this study, the introduction is very laborious, there are a lot of details which could be passed in the discussion section- eq- line 52-58. Also, bibliographical references are very old for prevalence of stones in gallbladder disease ( line 49- bibl 1- is from 1987)
- Line 97- the aim of the study should be more detailed- for example, evaluating the role of LERV in terms of efficiency, morbidity....
- In the methods section, it would be useful to define how pancreatitis was assessed - inflammatory markers? pancreatic enzymes? imaging? Were there other risk factors for pancreatitis besides gallstones? For example, diabetes, lipid changes? In Discution session pancreatitis is mentioned a little – about biochemical evaluation , line 340, but these details should be presented in the materials chapter because this complication is the most representative in this affection.
- Line 187- what does mean – short- term antibiotic therapy? one dose, one day, depending on clinical criteria or bile cultures?
- Was used per procedurally inhibition of pancreatic enzyme secretion? for example – octreotide
- Table 1- explanation of abbreviations are missing at the bottom of the table – Lap-chol? ERCP?
- I believe that more details are needed for the 5 patients who had stone recurrence: did they have a stent? Cholangiocarcinoma MRI? Was the stone de novo or... residual? Were there other risk factors? – line 248
- In Discution, should be mentioned the studies comparing the 2 approaches (line 297)
The bias and the limitations of the study are well presented in the discussion chapter.
Based on the obtained results, the authors conclude that the LERV technique is a safe procedure in selected cases and allows good clinical outcomes with lower incidence of clinically relevant post-operative ERCP pancreatitis. The conclusions are presented at the end of the Discussion chapter. I suggest that there be a separate chapter dedicated to the conclusions of this study.
The study have many bibliographic references from recent years, maybe it will be possible to give up the oldest representations .
Kind regards,
Author Response
- The abstract is a clear summary of the aim. There are abbreviations that are not explained on first appearance - for example, line 30 - CBD or line 32 – CDB. Also, there are situations in the text - line 92 for example - when an abbreviation is explained again (in this case, LERV) or line 177, or the abbreviation is given and then explained - for example, line 77 - EPBD (Endoscopic papillary balloon dilation). In the Abstract I think that instead of Discussion (line 40), it should be Conclusions of this study.
Comment: Thanks for the comment. As suggested, we have made corrections, and all abbreviations are now corrected. In the abstract Discussion is now changed with Conclusion
2 The introduction section should summarize the current state of the topic as well as clearly define the aim of the study. In this study, the introduction is very laborious, there are a lot of details which could be passed in the discussion section- eq- line 52-58. Also, bibliographical references are very old for prevalence of stones in gallbladder disease (line 49- bibl 1- is from 1987)
Comment: Thanks for the comment. As suggested, we have made corrections and in the revised version we have deleted line 52-58 and updated the reference cited.
3 Line 97- the aim of the study should be more detailed- for example, evaluating the role of LERV in terms of efficiency, morbidity....
Comment: Thanks for the comment. In the revised version we have included the reviewer suggestions (see line 126-129)
4 In the methods section, it would be useful to define how pancreatitis was assessed - inflammatory markers? pancreatic enzymes? imaging? Were there other risk factors for pancreatitis besides gallstones? For example, diabetes, lipid changes? In Discution session pancreatitis is mentioned a little – about biochemical evaluation, line 340, but these details should be presented in the materials chapter because this complication is the most representative in this affection.
Comment: Thanks for the comment. In the revised version we have added information about the diagnosis of PEP (see ref 19 and line 207-210). Regarding the second question, whether other risk factors of pancreatitis were evaluated we must say that the risk of pancreatitis in such patients is very rare. Indeed, diabetes mellitus and lipid changes have been associated with increased risk of pancreatitis in several studies, however, not all studies have found an association
5 Line 187- what does mean – short- term antibiotic therapy? One dose, one day, depending on clinical criteria or bile cultures?
Comment: Thanks for the comment. In the revised version we have changed “short- term antibiotic therapy” as “single shot..” that best explains the concept
6 Was used per procedurally inhibition of pancreatic enzyme secretion? for example – octreotide
Comment: Thanks for the comment. No inhibition of pancreatic enzyme secretion was used. There is no evidence that validates such a therapeutical approach
7 Table 1- explanation of abbreviations are missing at the bottom of the table – Lap-chol? ERCP?
Comment: Thanks for the comment. Overlap with rev #1. Correction done
8 I believe that more details are needed for the 5 patients who had stone recurrence: did they have a stent? Cholangiocarcinoma MRI? Was the stone de novo or... residual? Were there other risk factors? – line 248
Comment: Thanks for the comment. Reviewer 2 asks for more news about the 5 patients with stone recurrence. In the revised version, we added the requested information and specified what the treatment of the 5 cases was (Lines 345-347).
9 In Discussion, should be mentioned the studies comparing the 2 approaches (line 297)
Comment: Thanks for the comment. Overlap with rev #1. Correction done (see ref 20)
- Based on the results obtained, the authors conclude that the LERV technique is a safe procedure in selected cases and allows good clinical outcomes with lower incidence of clinically relevant post-operative ERCP pancreatitis. The conclusions are presented at the end of the Discussion chapter. I suggest that there be a separate chapter dedicated to the conclusions of this study.
Comment: Thanks for the comment. Overlap with rev #1. Correction done and a conclusion chapter is now included
- The study has many bibliographic references from recent years, maybe it will be possible to give up the oldest representations.
Comment: Thanks for the comment. Overlap with rev #1. Correction done and references are now updated (see ref 1, 3, 16, 19 and 20).

Round 2
Reviewer 1 Report
Comments and Suggestions for Authors
The authors have improved the manuscript after revision, but some points are still unresolved. These should be corrected before a favourable decision is made:
1. The Prisma flow diagram is not a table, it should be renamed as Figure 1. Since the authors did not use a PRISMA template, it cannot be titled a PRISMA flowchart. I would recommend renaming it 'Flowchart of the study'.
2. The authors were asked to provide more detail on the baseline characteristics (demographics) of the patients to verify that the two comparison groups were symmetrical. I disagree with their response, as providing detailed baseline data such as weight, height, BMI and comorbidities is critical to assessing whether the two comparison groups were symmetrical at baseline. While the authors argue that these parameters are not related to the outcomes of interest, they are important to understand potential confounding factors that could influence the results. The symmetry of the baseline characteristics ensures that the observed differences are due to the procedural approach and not the underlying differences between the groups. In addition, including this information would increase transparency, allow readers to assess the validity of the comparisons, and align the study with standard reporting practises, even for retrospective analyses.
3. The authors were asked to indicate next to each variable how the variable was expressed (as n (%) or median (IQR)), but they only added the IQR next to one variable. In each case, they should indicate next to each variable whether it is n (%) or median (IQR), or add the following note in the table legend: "Data were expressed as n (%) or median (IQR), as appropriate.
Author Response
Reviewer #1 – review round #2
- The Prisma flow diagram is not a table, it should be renamed as Figure 1. Since the authors did not use a PRISMA template, it cannot be titled a PRISMA flowchart. I would recommend renaming it 'Flowchart of the study'.
Comment: We have provided to change the previous “Prisma flowchart” in Figure 1 in “Diagnostic and therapeutical flow chart” as suggested.
- The authors were asked to provide more detail on the baseline characteristics (demographics) of the patients to verify that the two comparison groups were symmetrical. I disagree with their response, as providing detailed baseline data such as weight, height, BMI and comorbidities is critical to assessing whether the two comparison groups were symmetrical at baseline. While the authors argue that these parameters are not related to the outcomes of interest, they are important to understand potential confounding factors that could influence the results. The symmetry of the baseline characteristics ensures that the observed differences are due to the procedural approach and not the underlying differences between the groups. In addition, including this information would increase transparency, allow readers to assess the validity of the comparisons, and align the study with standard reporting practises, even for retrospective analyses.
Comment: As suggested we added to BMI values in Table 1. As you can see, the two groups are homogeneous. No differences in technical and post operative complications were found
- The authors were asked to indicate next to each variable how the variable was expressed (as n (%) or median (IQR)), but they only added the IQR next to one variable. In each case, they should indicate next to each variable whether it is n (%) or median (IQR), or add the following note in the table legend: "Data were expressed as n (%) or median (IQR), as appropriate.
Comment: Thanks for the comment. We have provided to better clarify when IQR o n (%) are expressed in table 1 and table 2.
